# Clinical Impact of Revised Ciprofloxacin Breakpoint in Patients with Urinary Tract Infections by *Enterobacteriaceae*

**DOI:** 10.3390/antibiotics10040469

**Published:** 2021-04-20

**Authors:** Ga Eun Park, Jae-Hoon Ko, Sun Young Cho, Hee Jae Huh, Jin Yang Baek, Kwan Soo Ko, Cheol-In Kang, Doo Ryeon Chung, Kyong Ran Peck

**Affiliations:** 1Medical Center, Division of Infectious Diseases, Department of Medicine, Konkuk University, Seoul 05030, Korea; kaeun84@gmail.com; 2Samsung Medical Center, Division of Infectious Diseases, Department of Medicine, Sungkyunkwan University School of Medicine, Seoul 06351, Korea; jaehoon.ko@samsung.com (J.-H.K.); sunyoung81.jo@samsung.com (S.Y.C.); heejae.huh@samsung.com (H.J.H.); collacin@gmail.com (C.-I.K.); iddrchung@gmail.com (D.R.C.); 3Asia Pacific Foundation for Infectious Diseases (APFID), Seoul 06326, Korea; jy34.baek@gmail.com (J.Y.B.); ksko@skku.edu (K.S.K.); 4Molecular and Cell Biology, Sungkyunkwan University School of Medicine, Seoul 05030, Korea

**Keywords:** *Enterobacteriaceae*, ciprofloxacin, urinary tract infections

## Abstract

In 2018, the Clinical and Laboratory Standards Institute (CLSI) revised ciprofloxacin (CIP)-susceptible breakpoint for *Enterobacteriaceae* from ≤1 μg/mL to ≤0.25 μg/mL, based on pharmacokinetic-pharmacodynamic (PK-PD) analysis. However, clinical data supporting the lowered CIP breakpoint are insufficient. This retrospective cohort study evaluated the clinical outcomes of patients with bacteremic urinary tract infections (UTIs) caused by *Enterobacteriaceae*, which were previously CIP-susceptible and changed to non-susceptible. Bacteremic UTIs caused by *Enterobacteriaceae* with CIP minimal inhibitory concentration (MIC) ≤ 1 μg/mL were screened, and then patients treated with CIP as a definitive treatment were finally included. Patients in CIP-non-susceptible group (MIC = 0.5 or 1 μg/mL) were compared with patients in CIP-susceptible group (MIC ≤ 0.25 μg/mL). Primary endpoints were recurrence of UTIs within 4 weeks and 90 days. A total of 334 patients were evaluated, including 282 of CIP-susceptible and 52 of CIP-non-susceptible. There were no significant differences in clinical outcomes between two groups. In multivariate analysis, CIP non-susceptibility was not associated with recurrence of UTIs. CIP non-susceptibility based on a revised CIP breakpoint, which was formerly susceptible, was not associated with poor clinical outcomes in bacteremic UTI patients were treated with CIP, similar to those of the susceptible group. Further evaluation is needed to guide the selection of definitive antibiotics for UTIs.

## 1. Introduction

Urinary tract infections (UTIs) are the most commonly encountered bacterial infection in the community [1,2]. In hospitals, UTIs are the second most common infection, accounting for nearly 25% of all infections [3]. Fluoroquinolones are one of most frequently prescribed classes of antibiotics in outpatient and inpatient settings. They have bactericidal action resulting from inhibition of topoisomerase II (DNA gyrase) and topoisomerase IV, which contributes to DNA replication, repair, and recombination. Ciprofloxacin (CIP) is a member of the fluoroquinolones group, which showed in vitro activity against both Gram negative bacilli and Gram positive bacteria, including *Enterobacteriaceae* [4]. CIP is one of the antibiotics most frequently prescribed for treating UTIs. This is due to the fact that CIP has a good bactericidal effect, clinical cure rates, high bioavailability exceeding 70%, and high concentrations in urine and kidney tissue [5]. However, resistance to quinolone has emerged all over the world following their widespread use. Three mechanisms of quinolone resistance have been reported: (1) chromosomal mutations in the genes encoding quinolone target enzymes (i.e., DNA gyrase or topoisomerase IV) that occur most often in a region referred to as the quinolone resistance determining region (QRDR); (2) the acquisition of plasmid-mediated quinolone resistance (PMQR) genes; and (3) chromosomal mutations that decrease intracellular concentration of quinolone by modification of the efflux pumps [6,7]. Recently, the Clinical and Laboratory Standards Institute (CLSI) revised CIP-susceptible breakpoint for *Enterobacteriaceae* from ≤1 μg/mL to ≤0.25 μg/mL, based on pharmacokinetic-pharmacodynamic (PK-PD) attainment analyses and in vitro bacteria eradication data [8,9,10,11,12]. However, PK-PD analysis and in vitro data may not accurately predict clinical outcomes for all infections, especially less severe infections such as UTIs. Moreover, clinical data supporting the lowered CLSI breakpoint are insufficient, and this revised MIC is also not based on the mechanism of quinolone resistance. In order to evaluate the clinical impact of revised CLSI breakpoint, we analyzed the clinical outcomes of patients receiving CIP for the treatment of UTIs with bacteremia caused by *Enterobacteriaceae* isolates that were previously CIP-susceptible and changed to non-susceptible. We compared those with CIP minimal inhibitory concentration (MIC) ≤ 0.25 g/mL and those with CIP MIC 0.5 and 1 g/mL.

## 2. Results

### 2.1. Study Populations, Baseline Characteristics, and Severity of Infections

A total of 334 patients were eligible. Fifty-two patients were in CIP-susceptible group and 282 patients were in CIP-non-susceptible group (Figure 1). Demographic characteristics are shown in Table 1. There were no significant differences between the two groups with regard to age, Charlson’s weighted index of comorbidity (CWIs), identified pathogens including extended-spectrum beta-lactamases (ESBL) producing organisms, treatment duration of antibiotics, and number of risk factors for recurrent UTIs. The most frequent etiologic organism was *Escherichia coli* (91%). CIP was empirically used as initial treatment regimen in 33% of 334 patients. The third generation cephalosporin (ceftriaxone or cefotaxime) was used in 40.4%. The proportion of patients receiving CIP as an empirical as well as a definitive treatment did not differ between CIP-susceptible and CIP-non-susceptible group (33% vs. 34.6%, *p* = 0.818, respectively). Female patients were more common in CIP-non-susceptible group than CIP-susceptible group (94.2% vs. 80.5%, *p* = 0.016, respectively). A lower proportion of patients in the CIP-susceptible group had liver disease (9.6% vs. 19.2%, *p* = 0.041, respectively) and renal disease (4.3% vs. 11.5%. *p* = 0.033, respectively) compared to patients in the CIP-non-susceptible group, and a higher proportion of patients in the CIP-susceptible group had solid cancer (16.3% vs. 5.8%, *p* = 0.044, respectively). The median value of pitt bacteremia score was not different between CIP-susceptible group and CIP-non-susceptible group, whereas standard deviation of median value was statistically significantly higher in CIP-susceptible group (1 [0–2] vs. 1 [0,1], *p* = 0.006, respectively). The proportion of patients with risk factors for recurrence of UTIs did not differ between two groups (26.6% vs. 26.9%, *p* = 0.961)

### 2.2. Clinical Outcomes of Patients with Enterobacteriaceae Bacteremic UTIs

The clinical outcomes of patients receiving CIP for the treatment of UTIs with *Enterobacteriaceae* bacteremia are presented in Table 2. Patients in the CIP-susceptible group remained hospitalized longer than patients in the CIP-non-susceptible group, but a statistically significant difference was not observed (6 (4–8) vs. 5 (4–6.75) days, *p* = 0.055). There were no significant differences in recurrence within 4 weeks (2.5% vs. 3.8%, *p* = 0.577, respectively), recurrence within 90-days (8.2% vs. 5.8%, *p* = 0.399, respectively), and all-cause mortality (1.1% vs. 1.9%, *p* = 0.494, respectively) between CIP-susceptible and CIP-non-susceptible group. Out of 334 patients, 282 patients underwent follow-up blood culture, and prolonged bacteremia more than one week were not observed in either group. Time to defervescence was longer in CIP-susceptible group than CIP-non-susceptible group (3 (2–4) vs. 2 (1–3) days, *p* = 0.003, respectively).

### 2.3. Multivariate Analysis of Relapse within 4 Weeks and 90 Days Recurrence

There were a few meaningful variables in univariate analysis of association between characteristics of patients and recurrence of UTIs within 4 weeks (Appendix A) and 90 days (Appendix A). A multivariate cox regression analysis was performed to adjust for the effects of potential confounding factors on recurrence of UTIs within 4 weeks and 90-days. Meaningful variables in univariate analysis were replaced with variables that could be represented. In addition to CIP non-susceptibility, age, sex, ESBL producing organism, CWIs, appropriate antibiotics, pitt bacteremia score, and risk factors for recurrent UTIs were included in the multivariable analysis. In the multivariate analysis, CIP non-susceptibility were not associated with recurrence within 4 weeks (Table 3) and 90 days (Table 4). Having a known risk factor for recurrence of UTIs was identified as a significant variable for recurrence of UTIs within 90 days (HR 2.370, 95% CI 1.050–5.351, *p* = 0.038).

### 2.4. A Subgroup of 111 Patients Treated with CIP Empirically as Well as Definitively

We also performed a subgroup analysis of patients empirically treated with CIP in CIP-susceptible group and CIP-non-susceptible group. Baseline characteristics of patients treated with CIP as an empirical treatment in CIP-susceptible group and CIP-non-susceptible group showed in Appendix A. Bacteremia due to *K.pneumoniae* was significantly more common in CIP-non-susceptible group than CIP-susceptible group (22.2% vs. 6.5%, *p* = 0.032, respectively). Except for this, there were no significant differences between two groups. Clinical outcomes were also not different between two groups (Appendix A). Comparison of the two groups was performed in the same methods as the primary analysis. CIP non-susceptibility was not associated with recurrence of UTIs within 4 weeks (HR 3.290, 95% CI 0.190–57.010, *p* = 0.413) (Appendix A) and 90 days (HR 0.945, 95% CI 0.092–9.682, *p* = 0.962) (Appendix A). Only age was associated with recurrence within 90 days (HR 0.926, 95% CI 0.862–0.994, *p* = 0.033).

## 3. Discussion

The susceptible breakpoints for antibiotics are essential for the clinicians to select an appropriate antibiotics for many types of infectious diseases. In fact, the work of establishing breakpoints conducted by the CLSI or the European Committee on Antimicrobial Susceptibility Testing (EUCAST) is basically based on PK and PD parameters of antibiotics [10]. Although the Clinical and Laboratory Standards Institute (CLSI) revised CIP-susceptible breakpoint for *Enterobacteriaceae* from ≤1 μg/mL to ≤0.25 μg/mL in 2018, clinical impact of the lowered CLSI breakpoint has not been evaluated. This retrospective cohort study showed that the CIP non-susceptibility according to revised CIP breakpoint, which was susceptible according to previous criteria, was not associated with recurrence of UTIs within 4 weeks and 90 days.

While there is growing evidence that inappropriate empirical antibiotic treatment may have a negative impact on clinical outcomes, its impact may also be confounded by other variables. Since *Enterobacteriaceae*, especially *E. coli*, is one of the main causative pathogens of both community- and hospital-acquired infection, the emergence of quinolone resistance of these pathogens is of great concern [13]. To date, many clinical studies have tried to evaluate the association between inadequate empirical treatment due to quinolone resistance and clinical impact in *Enterobacteriaceae* infections. Lautenbach et al. showed that quinolone resistance was an independent risk factor for mortality in patients with hospital acquired *Enterobacteriaceae* infection [13], whereas other studies have not. Peralta et al. showed that inadequate antibiotics were associated with worse prognosis in patients with *E. coli* bacteremia. However, bacteremia of urinary origin was associated with better prognosis [14]. Another previous report showed that discordant use of fluoroquinolone as an initial empirical treatment for bacteremic UTIs caused by *Enterobacteriaceae* could lead to worse clinical response and longer hospitalization. However, overall mortality and clinical cure rates were not associated with appropriateness of empirical antibiotics treatment [15]. Jeon J.H. et al. showed that that CIP is an appropriate choice for empirical treatment of UTIs and has no serious adverse outcomes, if it is adjusted appropriately, even for patients infected with CIP-resistant organisms [16]. As various studied had shown, bacteremic UTIs caused by *Enterobacteriaceae*, in particular, accounts for a large proportion of blood stream infections, which are often transient and have a better prognosis than other site of infections [14,15,16,17,18].

To identify the clinical outcomes according to revised CIP breakpoint in UTIs with *Enterobacteriaceae* bacteremia treated with CIP, we conducted a retrospective cohort study. CIP non-susceptibility according to revised CIP breakpoint was not associated with recurrence of UTIs. Considering that selection of the initial empirical antibiotics could affect the clinical outcomes, we also analyzed patients who treated with CIP as an empirical antibiotic. In this subgroup analysis, CIP non-susceptibility was not associated with recurrence of UTIs within 4 weeks and 90 days. One interesting point was that time to defervescence was longer in CIP-susceptible group than CIP-non-susceptible group. This means that statistically significant differences in time to defervescence did not correlate with other clinical outcomes.

There are possible explanations for reasons why CIP non-susceptibility according to the revised CLSI did not affect primary outcomes. First, the prognosis of UTIs caused by *Enterobacteriaceae* was better than that of other site of infections as mentioned above. Concentrations of quinolone in urine are generally high with a major renal route of elimination, and the flushing mechanism of the bladder expels pathogens. In case of the quinolone indicated for UTIs, CIP (30~50%) has significant renal excretion. The post antibiotic effect of quinolone also contributes to its therapeutic efficacy [17,18]. Therefore, lower MIC within non-susceptible range might not be a great concern in UTI patients. Second, less critically ill patients with relatively low CWIs and pitt bacteremia scores were included. Since patients with bacteremic UTIs due to *Enterobacteriaceae*, which were sensitive to CIP according to previous CLSI breakpoint, were included, it is likely that severely ill patients who had been repeatedly exposed to broad-spectrum antibiotics and hospital environment were excluded from the study. Third, when broth micro-dilution (BMD) method is used, it can reveal discrepancy in susceptibility results between the tested methods [19]. In other words, even if the CIP MIC by the automated antimicrobial susceptibility test (AST) method showed non-susceptible, there was a possibility that the CIP MIC would have been less than 0.5 μg/mL by the BMD method. The difference in clinical outcomes according to MIC by BMD method should be studied.

The currently revised CIP breakpoint for *Enterobacteriaceae* is based on PK-PD attainment analyses, which may not accurately predict clinical outcomes for all infections. This value is also not based on the mechanism of acquiring quinolone resistance genes. Quinolone resistance is principally caused by spontaneous mutation in the quinolone resistance determining region (QRDR). Multistep mutations of QRDR were responsible for high level resistance to quinolone [20,21]. However, the acquisition of plasmid-mediated quinolone resistance (PMQR) genes has been increasingly reported in *Enterobacteriaceae* species in most parts of the world [22,23]. The co-existence of mutations in QRDR and PMQR genes could occur, and the accumulation of these different mutations was associated with high level quinolone resistance [22,23,24,25,26,27,28,29]. There were also several reports that *Enterobacteriaceae* isolates, which are susceptible to quinolone, also had carried mutations in QRDR or PMQR genes [14,22,28,29,30]. There was a report that this led to them showing no clinical response when CIP resistance was acquired by the quinolone resistance gene, even when CIP MIC was about 0.25–0.5 μg/mL [30]. Clinicians should select appropriate antibiotics based on not only the results of AST but also the clinical response [14,15,16,31]. Further research should be performed on the association between acquiring quinolone resistance gene with CIP MIC and clinical outcomes.

Our study has several limitations. This is a retrospective observational study, and data collection was limited because data were gathered for only 10 years. Patients with recurrent UTIs might have sought treatment at another medical center or hospital, therefore some patients with recurrent UTIs may not have been included. Decision for choosing empirical antibiotics was made freely by physicians, resulting in major bias. The number of enrolled patients was small and showed low CWIs and Pitt bacteremia score. Our results cannot be applied to more severe UTIs presented with septic shock or cases with high risk of antibiotic resistance. Additionally, CIP MIC was measured by automated methods not by BMD method. As mentioned above, the MIC value by BMD may be different from the value by automated AST. However, our study has clinical significance reflecting real-world data where automated methods are usually used for AST.

## 4. Method

### 4.1. Study Designs and Patients

A retrospective cohort study was conducted from 2009 to 2018 in a single center, a 1950-bed tertiary care referral hospital in Seoul, South Korea. All adults patients (age > 18) diagnosed with bacteremic UTIs caused by *Enterobacteriaceae* considered susceptible to CIP by the prior breakpoints (CIP MIC ≤ 1 μg/mL) before 2018 CLSI revision were identified. Patients receiving CIP as a definitive treatment were included. Patients receiving more than two antibiotics as definitive treatment were excluded. Included patients were divided into CIP-susceptible (MIC ≤ 0.25 μg/mL) and CIP-non-susceptible groups (MIC = 0.5 or 1 μg/mL), according to the revised CLSI guideline in 2018.

### 4.2. Antimicrobial Susceptibility Testing

Species identification for all included patients was performed using the standard VITEK 2 identification card (bioMérieux Inc., Marcyl’Etoile, France) according to the CLSI susceptibility interpretative criteria [10].

### 4.3. Clinical Evaluation and Outcomes

Data were collected from electronic medical records. The following information were included: age, sex, the presence of underlying diseases or comorbid conditions, identified pathogens, types of antibiotics empirically administered, and treatment duration of antibiotics. The severity of underlying disease was estimated using CWIs. The severity of illness in bacteremia was assessed using the Pitt bacteremia score, which has been validated in several previous studies [32]. A history of urologic functional or structural abnormalities was collected as risk factors for recurrent UTIs, for example, neurogenic bladder, urinary incontinence, urolithiasis, kidney transplantation, or the presence of an indwelling foreign bodies (e.g., catheter, double J) [1]. Primary endpoints were recurrence of UTIs within 4 weeks and 90 days after administration of first antibiotics dose. Secondary endpoint included length of hospital stay from first antibiotics administration until discharge, all-cause mortality, prolonged bacteremia more than 1 week, and time to defervescence reflecting short-term clinical response. Patients who received CIP as an empirical antibiotics were analyzed to determine the primary endpoint as a subgroup analysis.

### 4.4. Definitions

UTIs were defined as a clinical syndrome characterized by costovertebral angle tenderness, fever, dysuric symptoms, pyuria, or bacteriuria [15]. Antimicrobial treatments were classified as empirical and definitive. Empirical treatment was defined as the antibiotics administrated before the final blood culture report, while definitive treatment was defined as the adjusted antibiotics based on the results of AST of blood culture isolates. Recurrence was defined as the presence of any symptoms or laboratory evidence of UTIs after discontinuation of antibiotic treatment.

### 4.5. Statistical Analysis

Differences between the CIP-susceptible and CIP-non-susceptible groups were compared using a Mann–Whitney U test for continuous variables and a chi-square test for categorical variables. The Cox proportional hazard model was used to examine the association of the CIP susceptibility with recurrences within 4 weeks and 90 days by adjusting for potential confounding factors. Variables with statistical significance in the univariate analysis or potential confounders were entered into the multivariable analysis. All *p*-values were two-tailed, and those <0.05 were considered statistically significant. IBM SPSS Statistics version 25.0 for Windows (IBM, Armonk, NY, USA) was used for all statistical analyses. The present study was approved by Institutional review board of Samsung medical center. Informed consent was waived since the electronic medical record was reviewed retrospectively with de-personalized identification number.

## 5. Conclusions

In a retrospective cohort study of UTI patients with *Enterobactericeae* bacteremia treated with CIP, increased CIP MIC was not associated with relapse within 4 weeks and recurrence within 90 days. This study suggests, even if the MIC of CIP is slightly elevated (0.5–1 g/mL) in *Enterobacteriaciae*, it could be possible to continue to use CIP in cases of bacteremic UTIs when patients show clinical response. However, our data cannot be extrapolated to patients with septic shock or severe infection other than UTIs.

## Figures and Tables

**Figure 1 antibiotics-10-00469-f001:**
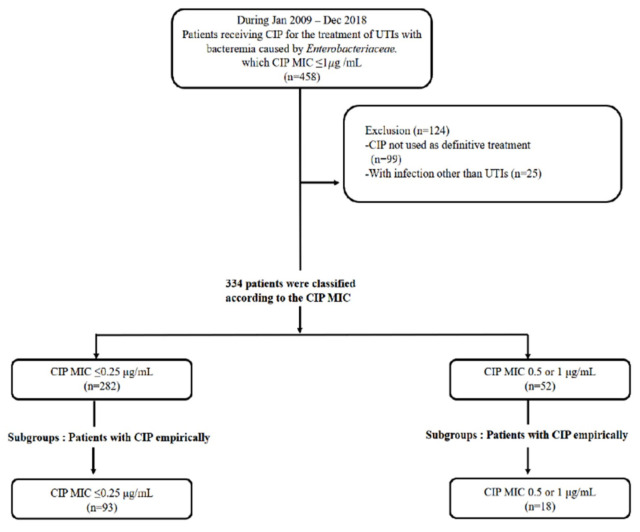
Flowchart of study population.

**Table 1 antibiotics-10-00469-t001:** Baseline characteristics of patients with bacteremic urinary tract infections caused by *Enterobacteriaceae* in ciprofloxacin-susceptible group and ciprofloxacin-non-susceptible group.

Variables	CIP-Susceptible MIC ≤ 0.25 (*n* = 282)	CIP-Non-Susceptible MIC = 0.5 or 1 (*n* = 52)	*p* Value
**Sex, female**	227 (80.5)	49 (94.2)	0.016
**Age (years)**	69 (55–78)	68.5 (51–77)	0.363
**Patients with any comorbidity**			
Diabetes mellitus	79 (28)	15 (28.8)	0.902
Cardiovascular disease	113 (40.1)	18 (34.6)	0.459
Respiratory disease	6 (2.1)	0 (0)	0.359
Liver disease	27 (9.6)	10 (19.2)	0.041
Renal disease	12 (4.3)	6 (11.5)	0.033
Neurologic disease	34 (12.1)	6 (11.5)	0.916
Solid cancer	46 (16.3)	3 (5.8)	0.048
Connective tissue disease	10 (3.5)	2 (3.8)	0.915
Hematologic disease	4 (1.4)	1 (1.9)	0.573
Solid organ transplantation	15 (5.3)	6 (11.5)	0.090
**CWI score**	1 (0–2)	1 (0–2)	0.558
**Pitt bacteremia score**	1 (0–2)	1 (0–1)	0.006
**Identified pathogen**			
*Escherichia coli*	256 (90.8)	48 (92.3)	0.723
*Klebsiella pneumoniae*	26 (9.2)	4 (7.7)	0.723
ESBL Producing organism	10 (3.5)	2 (3.8)	0.915
**Empirical antibiotics**			
Appropriate antibiotics	278 (98.6)	33 (63.5)	<0.001
Ciprofloxacin	93 (33)	18 (34.6)	0.818
Ceftriaxone	114 (40.3)	21 (40.4)	0.996
Piperacillin/tazobactam	47 (16.7)	9 (17.3)	0.909
Ertapenem	5 (1.8)	1 (1.9)	0.940
Others	23 (8.2)	3 (5.8)	0.399
**Treatment duration**	13 (11–17)	13 (10–15)	0.119
**Follow up duration**	107 (63–141.50)	100 (43.25–131.25)	0.401
**Risk factors for recurrence**	75 (26.6)	14 (26.9)	0.961
Previous UTIs history	36 (12.8)	8 (15.4)	0.608
Foreign body insertion	14 (5)	0 (0)	0.101
Urinary stone	12 (4.3)	0 (0)	0.130
Polycystic kidney disease	1 (0.4)	0 (0)	0.844
Urinary dysfunction	12 (4.3)	3 (5.8)	0.628
Kidney transplantation	13 (4.6)	6 (11.5)	0.047

Data are expressed as number (%) of patients or median (IQR). Abbreviations: CIP = ciprofloxacin; MIC = minimum inhibitory concentration; CWI = Charlson weighted index; ESBL = extended-spectrum beta-lactamases; UTIs = urinary tract infections.

**Table 2 antibiotics-10-00469-t002:** Comparison of clinical outcomes for patients with bacteremic urinary tract infections with *Enterobacteriaceae* bacteremia ciprofloxacin-susceptible group and ciprofloxacin-non-susceptible group.

Variables	CIP-Susceptible MIC ≤ 0.25 (*n* = 282)	CIP-Non-Susceptible MIC = 0.5 or 1 (*n* = 52)	*p* Value
Length of Hospital days	6 (4–8)	5 (4–6.75)	0.055
Recurrence within 4 weeks	7 (2.5)	2 (3.8)	0.577
Recurrence within 90 days	23 (8.2)	3 (5.8)	0.399
All-cause mortality	3 (1.1)	1 (1.9)	0.494
Prolonged bacteremia more than 1 week	0 (0)	0 (0)	1
Time to defervescence	3 (2–4)	2 (1–3)	0.003

Data are expressed as number (%) of patients or median (IQR). Abbreviations: CIP = ciprofloxacin; MIC = minimum inhibitory concentration.

**Table 3 antibiotics-10-00469-t003:** Multivariate analysis of association between characteristics of patients and recurrence of urinary tract infections within 4 weeks.

Variables	HR (95% CI)	*p* Value
Sex, female	1.391 (0.164–11.827)	0.763
Age	1.025 (0.979–1.073)	0.298
CIP, non-susceptible	0.888 (0.091–8.698)	0.918
ESBL producing organism	1.488 (0.099–22.392)	0.774
CWI score	0.904 (0.585–1.397)	0.649
Appropriate antibiotics	0.235 (0.017–3.240)	0.279
Pitt bacteremia score	1.267 (0.828–1.937)	0.275
Risk factors for recurrence	0.356 (0.044–2.886)	0.334

Abbreviations: CIP = ciprofloxacin; ESBL = extended-spectrum beta-lactamases; CWI = Charlson weighted index.

**Table 4 antibiotics-10-00469-t004:** Multivariate analysis of association between characteristics of patients and recurrence of urinary tract infections within 90 days.

Variables	HR (95% CI)	*p* Value
Sex, female	0.910 (0.331–2.504)	0.855
Age	0.979 (0.957–1.001)	0.066
CIP, non-susceptible	0.365 (0.075–1.791)	0.214
ESBL producing organism	1.538 (0.251–9.432)	0.642
CWI score	1.030 (0.854–1.241)	0.760
Appropriate antibiotics	0.316 (0.052–1.918)	0.211
Pitt bacteremia score	0.847 (0.578–1.242)	0.395
Risk factors for recurrence	2.370 (1.050–5.351)	0.038

Abbreviations: CIP = ciprofloxacin; ESBL = extended-spectrum beta-lactamases; CWI = Charlson weighted index.

## Data Availability

The data presented in this study are openly available in MDPI.

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
