# Peer review of "Clinical Impact of Revised Ciprofloxacin Breakpoint in Patients with Urinary Tract Infections by Enterobacteriaceae"

_antibiotics, 2021, doi:10.3390/antibiotics10040469_

Round 1

Reviewer 1 Report

Review of the manuscript antibiotics-1161082

  1. Recommendation

Minor revision

  1. Overview and general recommendation:

The study is exciting and currently considering the CLSI-revised ciprofloxacin susceptible breakpoint for Enterobacteriaceae. The topic of the research remains open to the future.

2.1. Major comments:

This study's major disadvantage is the limited control over data collection because data were gathered 10-years retrospectively.

 Title.The title is too long. The word „bacteremic” is redundant. „Ciprofloxacin” appears twice in the title.

Introduction. The introduction part is too short. There is a need for more data regarding ciprofloxacin in terms of the type of drug (class, chemical structure), essential pharmacokinetics data, antibacterial activity, and therapy use. Relevant and updated data on bacterial resistance to ciprofloxacin are missing. The paragraph from 292 to 193-line (overview to resistance to antibacterial quinolones) is more appropriate for the introduction.

Line 38.

Ciprofloxacin and levofloxacin are antibacterial fluorinated quinolones (with accepted generic name fluoroquinolones). The term quinolone is too general (many compounds belong to this chemical group, but not all have an antibacterial effect). Please review this term all over the manuscript.

The two compounds are classified into different generations (ciprofloxacin – in the second generation and levofloxacin - in the third generation of antibacterial quinolones) with varying activity spectrum. Complete this specification in the manuscript.

2.2. Minor comments:

Explain the abbreviations BMD and AST on first use in the manuscript.

English writing needs to be improved. Some examples are

Line 20. Add a comma after <Enterobacteriaceae> (also use italic style)

Line 22. Add a comma after <screened>.

Line 25. Add a comma after <evaluated>.

Line 28. Replace <were> with <was>.

Line 30. Considers adding <the> before <selection>.

Use the generic Gram-negative and Gram-positive name (with a capital letter) all over the manuscript.

...and so on.

Author Response

Point 1: This study's major disadvantage is the limited control over data collection because data were gathered 10-years retrospectively

Response 1:

We added the major comment you pointed out to the limitation of the paper. We added the sentence following

: Data collection was limited because data were gathered only 10 years (Page 7, Line 221-222)

Point 2: The title is too long. The word bacteremic is redundant. Ciprofloxacin appears twice in the title

Response 2: We’ve changed the title to be more concise as follows according to your comment    

: “Clinical impact of revised ciprofloxacin breakpoint in patients with urinary tract infections by Enterobacteriaceae”

Point 3: The introduction part is too short. There is a need for more data regarding ciprofloxacin in terms of the type of drug (class, chemical structure), essential pharmacokinetics data, antibacterial activity, and therapy use. Relevant and updated data on bacterial resistance to ciprofloxacin are missing. The paragraph from 292 to 293-line (overview to resistance to antibacterial quinolones) is more appropriate for the introduction.

Response 3: We reinforced the introduction part as you recommended. The highlight line is the reinforced part (Page 1-2, Line 33-62)

: Urinary tract infections (UTIs) is the most commonly encountered bacterial infection in the community. In hospitals, UTIs are the second most common infection, accounting for nearly 25% of all infections. Flouroquinolones are one of most frequently prescribed classes of antibiotics in outpatient and inpatient settings. It has bactericidal action resulted from inhibition of topoisomerase II (DNA gyrase) and topoisomerase IV, which contribute to DNA replication, repair, and recombination. Ciprofloxacin (CIP) is member of flouroquinolones group which showed in vitro activity against both Gram negative bacilli and Gram positive bacteria, including Enterobacteriaceae. CIP is one of the antibiotics most frequently prescribed for treating UTIs. This is due to the fact that CIP have a good bactericidal effect, clinical cure rates, high bioavailability exceeding 70% and show high concentrations in urine and kidney tissue. However, resistance to quinolone has emerged all over the world following their widespread use. Three mechanisms of quinolone resistance have been reported ; 1) chromosomal mutations in the genes encoding quinolone target enzymes (i.e. DNA gyrase or topoisomerase IV) that occur most often in a region referred to as the quinolone resistance determining region (QRDR); 2) the acquisition of plasmid-mediated quinolone resistance (PMQR) genes; 3) chromosomal mutations that decrease intracellular concentration of quinolone by modification of the efflux pumps. Recently, the Clinical and Laboratory Standards Institute (CLSI) revised CIP susceptible breakpoint for Enterobacteriaceae from≤1μg/mL to≤0.25μg/mL, based on pharmacokinetic-pharmacodynamic (PK-PD) attainment analyses and in vitro bacteria eradication data. However, PK-PD analysis and in vitro data may not accurately predict clinical outcomes for all infections, especially less severe infections such as UTIs. Moreover, clinical data supporting the lowered CLSI breakpoint is insufficient and this revised MIC is also not based on the mechanism of quinolone resistance. In order to evaluate the clinical impact of revised CLSI breakpoint, we analyzed the clinical outcomes of patients receiving CIP for the treatment of UTIs with bacteremia caused by Enterobacte-riaceae isolates which were previously CIP susceptible and changed to non-susceptible. We compared those with CIP minimal inhibitory concentration (MIC)≤0.25μg/ml and those with CIP MIC 0.5 and 1μg/ml.

Point 4: Line 38 - Ciprofloxacin and levofloxacin are antibacterial fluorinated quinolones (with accepted generic name fluoroquinolones). The term quinolone is too general (many compounds belong to this chemical group, but not all have an antibacterial effect). Please review this term all over the manuscript.

The two compounds are classified into different generations (ciprofloxacin in the second generation and levofloxacin - in the third generation of antibacterial quinolones) with varying activity spectrum. Complete this specification in the manuscript.

Response 4 : Since this paper was investigated only in the patients treated with ciprofloxacin, it was decided to delete the mention of levofloxacin, and the term for quinolone was changed to fluoroquinolone.

(Page 1, Line 39-40)

Point 5: Explain the abbreviations BMD and AST on first use in the manuscript

Response 5 : We explained the abbreviations BMD and AST on first use in the manuscript

(Page 7, Line198-200)

Point 6: Line 20- Add a comma after <Enterobacteriaceae> (also use italic style)

Response 6 : We added a comma after <Enterobacteriaceae> and changed to italic style

Point 7: Line 22- Add a comma after <screened>

Response 7 : The two sentences are connected with and, so I don't think it needs a comma. A few words have been added to clarify the meaning of the sentence.

: Bacteremic UTIs caused by Enterobacteriaceae with CIP minimal inhibitory concentration (MIC)≤1μg/mL were screened and then patients treated with CIP as a definitive treatment were finally included (Page 1 Line 21-22)

Point 8: Line 25. Add a comma after <evaluated>

Response 8 : Since it is a sentence with a qualifying phrase “including 282 of CIP susceptible and 52 of CIP non-susceptible” at the end, it is unlikely that a comma is required after “evaluated”.

A total of 334 patients were evaluated including 282 of CIP susceptible and 52 of CIP non-susceptible (Page 1, Line 24)

Point 9: Line 28- Replace <were> with <was>.

Response 9 : We replaced <were> with <was> (Page 1, Line 27)

Point 10: Line 30- Considers adding <the> before <selection>

Response 10 : We added <the> before <selection> (Page 1, Line 29)

Point 11: Use the generic Gram-negative and Gram-positive name (with a capital letter) all over the manuscript.

Response 11 : We used the generic Gram-negative and Gram-positive name with a capital letter

Reviewer 2 Report

The paper fills a good gap in the literature, with a relevant question that should be answered. There are a few things that I recommend changing:

  • There are quite a few language/grammar errors; please go through the paper again.
  • Some abbreviations are not defined (such as on page 2 line 58, both CWI and ESBL have not been previously defined).

Unfortunately the number of patients is low, but I am glad you address this as a limitation.

The other limitation may be as this is retrospective, recurrent patients may have sought treatment at another medical center or hospital, so all recurrent patients may not be included. I believe this needs to be addressed.

Author Response

Point 1: There are quite a few language/grammar errors; please go through the paper again.

Response 1:

We checked the grammar errors again and we will check once more using the English proofing system before final submission

Point 2: Some abbreviations are not defined (such as on page 2 line 58, both CWI and ESBL have not been previously defined).

Response 2: We defined abbreviations of CWIs, ESBL etc. (Page 2, Line 68-69)

Point 3: Unfortunately the number of patients is low, but I am glad you address this as a limitation.

The other limitation may be as this is retrospective, recurrent patients may have sought treatment at another medical center or hospital, so all recurrent patients may not be included. I believe this needs to be addressed.

Response 3: We added the sentence following :

Patients with recurrent UTIs might have sought treatment at another medical center or hospital, therefore all patient with recurrent UTIs may not be included. (Page 7, line 222-224)